# What Works to Improve Nutrition and Food Sustainability across the First 2000 Days of Life: A Rapid Review

**DOI:** 10.3390/nu14040731

**Published:** 2022-02-09

**Authors:** Rachel Laws, Megan Adam, Emma Esdaile, Penelope Love, Karen J. Campbell

**Affiliations:** Institute for Physical Activity and Nutrition, School of Exercise and Nutrition Science, Deakin University, Geelong, VIC 3220, Australia; megan.adam@deakin.edu.au (M.A.); emma.esdaile@sydney.edu.au (E.E.); penny.love@deakin.edu.au (P.L.); karen.campbell@deakin.edu.au (K.J.C.)

**Keywords:** nutrition, food, infants, toddler, child obesity prevention, sustainability, rapid review

## Abstract

Informed by the Innocenti framework, this rapid review of systematic reviews (*n* = 60) and sentinel grey literature (*n* = 27) synthesises the evidence of what works to improve nutrition and food sustainability across the first 2000 days. Most systematic reviews focused on interventions targeting the behaviour of parents and caregivers (*n* = 49), with fewer reviews focusing on the personal (*n* = 7) and external (*n* = 4) food environments. No reviews focused on food supply-chain activities. Most reviews were rated as critically low (*n* = 28, 47%) or low (*n* = 21, 35%) quality using AMSTAR 2. Evidence supports the effectiveness of multi-component breastfeeding interventions, interventions delivered in home and child-care settings, particularly when involving parents, interactive skill building and repeated exposure to vegetables. Food vouchers and access to local farmers markets and community gardens have potential for improving access and availability to healthier foods, while evidence supports interventions improving the external food environment, including fiscal strategies such as the SSB tax, restrictions on marketing and advertising of discretionary products and improved food labelling. Overall, this review highlights the importance of action across a range of settings and sectors at the international, national and local levels to improve young children’s diets.

## 1. Introduction

A growing body of evidence indicates that the first 2000 days, from conception to age 5 years, is a critical time for shaping lifelong nutrition and subsequent health, development and wellbeing outcomes [1]. Diet quality and overweight and obesity have been shown to track from early childhood to later childhood and into adulthood [2]. Excessive and rapid weight gain in infancy has been linked to obesity in later life, as well as to a number of risk factors for cardiovascular disease [3,4]. For example, Zheng and colleagues [5] found that children experiencing rapid weight gain during the first two years of life are nearly four times more likely to be overweight or obese later in life.

Despite the known benefits of early life nutrition in promoting optimal child development, few Australian children meet nutrition guidelines. In 2017–2018, only 29% of infants were exclusively breastfed to 6 months of age [6] despite well-established benefits for both mother and child [7]. Further, evidence highlights that children are less likely to meet recommended dietary guidelines as they progress through the first five years of life. For example, research from the 2014–2015 Australian National Health Survey shows that, while still low, 20.2% of children aged 2–3 years met the recommended 2.5 serves of vegetables each day, but only 3.3% of children aged 4–5 years met the recommended 4.5 serves [8]. Further, almost all (96.8%) 2–3-year-olds and 98.9% of 4–8-year-olds consumed more than the recommended amount of discretionary foods high in fat, salt and sugar [9]. The consequences of this suboptimal diet are apparent, with 25% of Australian children overweight or obese by 2–4 years of age [10], an increase from 20% in 2015 [11]. This reflects worldwide trends, with a global estimate of 38 million children aged under 5 years of age overweight or obese in 2020 [12].

Optimising nutrition early in life is not only critical for human health but also for the health of the planet. Breastfeeding and establishing dietary patterns rich in plant-based foods and low in processed foods and animal products have been identified as important for both health and environmental sustainability [13]. As a result, there is increasing focus shifting towards healthy and sustainable diets using a food-systems approach. Food systems refer to the interdependent processes involved in getting food from the farm to the table [14]. Such an approach provides a useful lens through which to consider the multitude of potential influences and points to intervene to improve the diets of children as well as food sustainability. A food-systems approach should engage actors at all levels to reshape the food system and ensure that it delivers healthy, affordable, accessible and sustainable diets to all children [14].

There is a lack of evidence synthesis to guide practice and policy about what works to improve nutrition and food sustainability across the first 2000 days of life using a food-systems approach. Previous reviews in young children have focused on systematic reviews of interventions to improve specific nutrition outcomes, for example, fruit and vegetable intakes [15,16,17,18], breastfeeding [19], reducing discretionary foods [20]. Other reviews have focused on interventions delivered in particular settings such as child care [21,22] and the home [23] or targeting specific groups, such as parents [24,25,26], socioeconomically disadvantaged families [27] and Indigenous populations [28]. To our knowledge, no reviews have synthesized the evidence of the effectiveness of early life nutrition interventions across multiple domains taking a food-system approach and considering the issue of food sustainability.

Thus, to account for the broad range of influences on children’s diets we undertook a rapid review to synthesise the evidence for what works to improve nutrition and food sustainability across the first 2000 days of life using a food-systems approach. The findings of this rapid review are expected to provide broad guidance for practitioners and policy makers about ‘best buys’ and areas in which to invest to promote healthy and sustainable diets across the first 2000 days of life.

## 2. Materials and Methods

### 2.1. Study Overview and Conceptual Framework

This study consisted of a review of existing systematic reviews using a rapid-review methodology consistent with Cochrane collaboration guidance [29]. The review was part of a broader project commissioned by the Victorian Health Promotion Foundation to guide investments to improve nutrition across the first 2000 days of life. Other components of the project included a mapping of existing initiatives, policies and programs, in addition to key stakeholder interviews exploring opportunities to improve nutrition and food sustainability in the first 2000 days of life within Victoria, Australia. The review question and protocol were developed in close collaboration with the commissioning agency.

The UNICEF Innocenti Framework [14] was used as a conceptual framework to guide the review as it takes a broad food-systems approach to considering the influences on young children’s diets. The framework outlines four broad determinants of children’s diets: (1) *food supply chains,* which include the actors and activities that take food from ‘farm to table’—including production, storage, processing, distribution, packaging, retail and markets, and waste disposal; (2) *external food environments,* which include components such as price and availability of food, marketing and advertising, properties of food retail/service outlets and products; (3) *personal food environments,* which represent the individual and household factors that inform why people choose to procure the foods that they do; and (4) *behaviours of caregivers and children,* which refers to the food procurement, preparation, supervision, and eating practices of children and their caregivers [14].

### 2.2. Search Strategy

A search strategy was developed using the PICOTT method [30] (Table 1) to identify key concepts relevant to the review question. Given the short timeframe for conducting the review (3 months) and the large number of primary studies and systematic reviews, we limited this evidence synthesis to systematic reviews and meta-analyses published in English between January 2010 and July 2020. The search was rerun in June 2021 to identify any additionally published studies for inclusion in this rapid review.

Search terms reflecting PICOTT concepts were developed in collaboration with the commissioning agency and the university librarian. These comprised a series of key words (Appendix A) used to search titles and abstracts of articles in selected databases, namely EBSCO Host—CINAHL; Medline; Global Health; ERIC; Embase; Cochrane Database of Systematic Reviews; The Campbell Collaboration. Key words were adjusted to accommodate the specific coding requirements of each database, but this was done in a way that maintained integrity of the search parameters and to ensure a consistent search methodology was applied across all the selected databases.

### 2.3. Study Selection

All records from the initial database searches were transferred to reference management software (Endnote) and duplicates removed. Initial screening was conducted by one author (MA) on the basis of titles and abstracts to remove papers that were not systematic reviews or meta-analyses or were deemed irrelevant according to the PICOTT criteria (Table 1). Full text screening of remaining articles was completed by two authors (M.A. and E.E.) to determine eligibility for inclusion.

### 2.4. Quality Assessment

The methodological quality of included systematic reviews was independently assessed by two reviewers (M.A. and E.E.) using the AMSTAR 2 rating tool [31]. Differences in quality ratings were resolved by discussion. AMSTAR 2 consists of 16 items across 7 critical domains, namely the registration of the review protocol; adequacy of the literature search; justification for exclusion of each potentially relevant study; risk of bias from individual studies being included in the review; appropriateness of meta-analytical methods; consideration of risk of bias when interpreting the results of the review; presence and likely impact of publication bias [31]. Quality ratings were deemed high, moderate, low and critically low based on the number of identified weaknesses across the seven critical domains. A review was considered high quality if it had no more than one non-critical weakness, moderate quality if it had more than one non-critical weakness, low quality if it had one critical flaw (with or without non-critical weakness) and very low quality if it had more than one critical flaw (with or without non-critical weakness). It is important to note that the AMSTAR 2 tool assesses the quality of the systematic reviews, not the individual studies included in the reviews.

### 2.5. Grey Literature

It is well understood that a search that accesses predominantly peer-reviewed publications searchable in commercial databases will likely omit key information and result in publication bias [32]. Therefore, a grey literature search was conducted by one author (M.A.) using the advanced search functions of Google. Within the single search box of Google, nesting brackets were used to search multiple keywords (Appendix A). This search was limited to specific sites or domains to increase the relevance and reliability of the sources, as follows: .gov, .gov.au, .who.int, .edu, .edu.au. To control the quantity of items returned, the search was limited to the first 200 results as Google returns the most relevant documents first. The same inclusion and exclusion criteria as for the database sourced literature were applied. The Google searches were supplemented with a search of selected websites (for example The National Library for Public Health (NLPH), National Institute for Health and Clinical Evidence (NICE); World Health Organization (WHO) publications and reports).

### 2.6. Data Extraction and Synthesis

In line with the Cochrane rapid-review guidelines [29], one author (M.A.) extracted relevant information from each study (author, publication date, title, aim, settings and population, intervention types, year of studies if reported, countries of included studies, review findings and conclusion) and this was cross-checked by a second author (E.E.) for correctness and completion. The findings across reviews were synthesized narratively, organized by Innocenti domains and results grouped by intervention type, population and outcomes.

## 3. Results

A total of 7771 records were identified from database searches. After removing duplicates, screening of titles and abstract and full text assessment, a final sample of 60 reviews was included (Figure 1), along with 23 grey literature reports (Table 2).

### 3.1. Characteristics and Quality of Included Reviews

Of the 60 reviews, the majority (*n* = 49) focused on the behaviours of caregivers and children, with a small number focusing on personal food environments (*n* = 4) and external food environments (*n* = 7). No systematic reviews or meta-analyses for the food supply determinant of the Innocenti framework were identified (Table 3). The majority of reviews were rated as critically low (*n* = 28, 47%) or low (*n* = 21, 35%) in quality, with no studies rated as moderate and 11 (18%) rated as high quality (Table 2).

### 3.2. Behaviour of Caregivers and Children

Of the 60 reviews, 49 focused on the behaviour of caregivers and children, with most rated as low (17/49, 35%) or critically low (23/49, 47%) in quality (Table 2). The majority (30/49, 61%) focused on breastfeeding interventions, with a smaller number focusing on nutrition interventions targeting parents (*n* = 11), fruit and vegetable consumption (*n* = 5) and nutrition interventions within the Early Childhood Education and Care (ECEC) setting (*n* = 2). A summary of the effectiveness of various interventions in this domain is provided in Table 3 and summarized below.

#### 3.2.1. Breastfeeding Interventions

There was evidence from one high-quality review [34], indicating that breastfeeding education, support and counselling (both peer and health professional) improved breastfeeding initiation. The evidence regarding the benefit of antenatal breastfeeding education alone was insufficient [33]. Reviews supported the effectiveness of education and support in promoting exclusive breastfeeding [19,35,36,37,43,44,70] as well as the benefits of peer and health professional counselling (including group approaches) in improving initiation, duration and exclusivity [35,36,38,39,40,41].

Findings need to be interpreted with caution given that they come from low-quality reviews. The key elements of effective breastfeeding interventions identified from these reviews included:Provision of support across the antenatal and postnatal period;Mother–infant skin-to-skin contact after birth;Long duration of postpartum support (at least 4 contacts postnatally), with a focus on health professional contact in the first month postpartum;Provision of informative, social, emotional and instructional support;A focus on breastfeeding self-efficacy;Combining both educational and counselling approaches (multi-component);Involving health professionals, including combining peer support with leadership of a health professional or lactation consultant;Interventions concurrently delivered in a combination of settings;Involving partners/fathers.

While the delivery of breastfeeding interventions via telephone, eHealth and mHealth is gaining popularity, the results of reviews in this area are mixed. One high-quality review [46] concluded that targeted client communication using mobile phones (e.g., text messaging, voice calls and apps with instant messaging) may increase exclusive breastfeeding in settings where rates of exclusive breastfeeding are less common but had little effect when population rates of breastfeeding are high. Another high-quality review [45] reported inconsistent results from trials offering breastfeeding telephone support (mainly in the form of voice calls from health professionals). Finally, one review [47] (critically low quality), focused on internet support for breastfeeding and identified only one study (reporting positive results) for inclusion. This highlights the need for further high-quality evidence to support breastfeeding interventions delivered via telephone, eHealth and mHealth, the need for which is even more evident in light of the use of these modes of service delivery during COVID.

There was clear evidence that tailored breastfeeding support is required to meet the specific needs of priority population groups including adolescents, women with overweight/obesity, and families who face systemic inequities such as low-income, racial and ethnic minority communities. Further research is required, however, to identify the key components of effective interventions targeting these specific groups [48,49,50,51,52]. There was evidence that interventions targeting pregnant Indigenous women can improve rates of breastfeeding initiation and duration, with successful interventions typically multi-component and involving individual counselling delivered by Indigenous workers or peers. We use the term Indigenous here to refer to First Nations communities internationally as the studies included in this review come from multiple countries. These findings need to be interpreted with caution, however, as they came from just one review rated as low quality [28].

At the settings level, there was strong endorsement by WHO [71] for policy and programs that support breastfeeding in healthcare services and workplaces. There was some evidence to support the effectiveness of workplace interventions [57,58] and the Breastfeeding Friendly Health Initiative (BFHI) [53,54], including positive impacts amongst minority women [52]. There was also evidence suggesting that interventions delivered in a combination of settings (health systems, home, family, community) improved breastfeeding outcomes [36]. However, further research is required before firm conclusions can be drawn given that these findings are based on the results of reviews with low-to-moderate quality ratings. There was insufficient good quality evidence [55,56] about whether the provision of breastfeeding training to healthcare staff improves knowledge of and/or attitudes to breastfeeding of healthcare staff, compliance with BFHI or compliance with the International Code of Marketing of Breastmilk Substitutes.

At a macro level, the review by Segura-Perez [52], rated as high in quality, found that the United States federal and state policies supporting breastfeeding had a positive impact on breastfeeding practices amongst minority women. These laws included workplace laws (requiring the provision of lactation breaks and private space for expressing or breastfeeding) and the Affordable Care Act (ACA) mandating coverage of lactation support services. This positive impact was also seen in a review [36] looking at interventions in different settings, including workplace policies.

#### 3.2.2. Parent and Early Childhood Education and Care Focused Nutrition Interventions

After the transition to solid food, a child’s diet remains heavily influenced by the behaviour of caregivers, including the caregiver’s diet. The evidence regarding the impact of nutrition education delivered during pregnancy on maternal diet was mixed in interventions targeting mainstream [59,60], adolescent [61] and Indigenous populations [28]. Similarly, there was mixed evidence about the efficacy of parent-focused interventions on child dietary outcomes, with most reviews in this area being of low quality [59,62,63,64,65,66]. One high-quality review [62] concluded that there is insufficient good quality evidence to support the effectiveness of family-focused nutrition education interventions delivered in the first year of life in reducing the risk of child overweight and obesity.

Across reviews, the features of effective parent-focused interventions included:Clear, simple nutrition messages;Theory-based interventions;Direct engagement of parents (e.g., interactive sessions with parent skills training) rather than passive information provision;Responsive feeding education;Engaging both parent and child (for preschool children including ‘hands on’ experiences);Higher contact time and a range of intervention strategies;Using a parent-preferred delivery mode to increase engagement (e.g., using videos rather than written materials for adolescents).

A grey literature [67] report concluded that digital interventions targeting parents show particular promise in improving both parent and child dietary intakes [67], although it is unclear how applicable this evidence is to parents of young children. Credible content that includes both informational and interactive components is key to engagement and use of digital interventions [67]. As these findings have arisen from low-quality reviews, cautious interpretation is recommended.

There was strong evidence from high-quality reviews [15,68] that repeated exposure to vegetables is an effective strategy for increasing vegetable intake both in parent-focused interventions and those delivered in early childhood education and care (ECEC) settings. There was also strong evidence [15] for multi-component interventions (e.g., combining parent education with changes in ECEC policy) in improving fruit and vegetable intake in young children. Reviews also identified ECEC as an effective setting for delivery of behavioural nutrition and social-marketing interventions to children and parents, although this evidence comes from low-quality reviews [21,22] and hence caution is required before firm conclusions can be drawn.

### 3.3. Personal food Environments

Four of the 60 reviews focused on aspects of the personal food environment, specifically the impact of food supplementation and food vouchers on child nutrition, as well as the impact of the availability and access to farmers markets and gardening interventions on fruit and vegetable consumption. Only one of the four reviews in this domain was rated as high quality [72] (Table 2). A summary of the effectiveness of various interventions in this domain is provided in Table 4 and summarized below.

#### 3.3.1. Supplementation

One high-quality systematic review and meta-analysis assessed the impact of supplementary feeding interventions on the physical and psychosocial health of children at risk of undernutrition living in disadvantage [72]. Supplementary feeding consisted of additional meals, snacks and/or drinks provided in preschool, day care, or community settings as well as take-home or home-delivered food. This review supported the hypothesis that supplementary feeding programs can improve nutrition and growth of children aged 3 months to 5 years, particularly in low- and middle-income countries. Interventions were more effective for disadvantaged children under 2 years, and when delivered outside of the home under supervised conditions. Impacts on promoting healthy weight in disadvantaged children in high-income countries varied and requires further research.

#### 3.3.2. Food Vouchers

Two systematic reviews [73,74], both with a quality rating of critically low, examined the impact of changes to the U.S. Special Supplemental Nutrition Program for Women, Infants, and Children (WIC) food-package policies introduced in 2009. The changes aimed to increase access to and consumption of fruit, vegetables, whole grains and low-fat milk amongst WIC participants who acquired food from approved retail stores using provided cheques, vouchers or electronic funds transfer facilities. Both reviews supported improved availability of healthier foods in approved stores post the 2009 changes. Schultz et al. [73] also reported increase in purchases of healthier foods and improved dietary intake of WIC participants; however, the impact on breastfeeding outcomes was mixed.

In line with these findings, a grey literature evidence summary [75] on equity in early childhood development reported that food subsidy and voucher programs offer potential for improving nutrition related inequities among pregnant women and families with young children, with some evidence of the positive impact on children’s diets and health outcomes. Further, a Healthy Food America Research Report [76] provided further evidence from the U.S. supporting the usefulness of vouchers in supporting local agriculture, in turn generating and sustaining regional infrastructure for production, distribution and processing of locally grown produce [27]. These findings need to be interpreted with caution due to the low quality of the published reviews and lack of peer-review evidence for the grey literature findings.

#### 3.3.3. Farmers Markets

The review by Zhang [74] provided emerging evidence of a positive association between farmer-to-consumer sales and fruit and vegetable purchases and/or consumption amongst WIC participants [74]. These findings were further supported by a review of the grey literature that found that supporting and increasing the presence of local, healthy food outlets such as farmers markets is likely to enhance the purchase of healthy foods while also supporting local agriculture and the regional economy [27,77,78,79]. Due to the low-quality rating of this evidence, these findings need to be confirmed in future research.

#### 3.3.4. Gardening Interventions

One review of gardening interventions in schools, community or after-school settings amongst children 2 to 15 years [18] showed a small but positive impact on children’s fruit and vegetable consumption. The confidence in these findings is low given the low quality of the review and the methodological limitations of many of the included primary studies. Future research in this area should focus on long-term changes in fruit and vegetable consumption; the influence of parental components of gardening-based interventions on fruit and vegetable consumption of participating children; the effects of duration and intensity of programs; and the use of age-specific curricula on program outcomes.

### 3.4. External Food Environments

Seven of the 60 reviews focused on aspects of the external food environment (Table 2). One high-quality review focused on the effectiveness of various environmental interventions on reducing sugar-sweetened beverage (SSB) consumption [80] in the general population. Another review of low quality [81] focused on the effectiveness of interventions in reducing SSB consumption specifically in 0–5-year-olds. Other reviews in this domain focused on the influence of marketing and advertising to children, with the quality of these reviews rated as low or critically low. The effectiveness of various interventions in this domain is synthesized below (Table 5).

#### 3.4.1. Physical Food Environment

Reviews focusing on SSBs [80,81] found that reducing access to SSBs and improving access to healthier alternatives both in the home [80] and other settings young children occupy [81] is effective in reducing SSB consumption. However, the effectiveness of these types of interventions in schools was mixed as was the inclusion of a healthier default beverage in children’s menus in restaurants and cafes [80].

#### 3.4.2. Fiscal/Economic Environment

The high-quality review by Von Philipson et al. [80] demonstrated the effectiveness of a number of fiscal strategies in decreasing SSB sales/consumption, including price increases/SSB tax and the use of food-voucher programs that specifically restricted SSB purchases while incentivizing purchasing fruit and vegetables. In contrast, food voucher programs that did not specifically restrict SSBs purchases had mixed effects on SSB sales. Similarly, offering price discounts on low-calorie beverages via supermarket loyalty cards had mixed results on SSB consumption patterns.

#### 3.4.3. Marketing and Advertising

There was consistent evidence across a number of reviews for the negative impact of unhealthy food marketing and advertising on children [20,82,83,84,85]. However, the quality of these reviews was low or critically low, and hence the findings need to be interpreted with some caution. A systematic review and meta-analysis (low quality) by Russell et al. [83] reported that in experimental studies children exposed to unhealthy food advertising on TV and advergames (interactive online games designed to deliver marketing messages) on digital devices consumed an average 60.0 kcal and 53.2 kcal, respectively, more than children exposed to non-food advertising. Non-experimental studies revealed that exposure to unhealthy food advertising on television was positively associated with, and predictive of, dietary intake in children. These findings are in line with those reported by Smith et al. [82], who found that food-marketing techniques that used advertising of unhealthy food through television, movies and product packaging were likely to be particularly effective. This is further supported by the review undertaken by Kraak and colleagues [84] that found that advertising with branding using familiar media characters influences children’s food preferences, choices and intake, and this effect was reported to be greater for energy-dense and nutrient-poor foods than for fruits or vegetables. In line with this, a review of the impact of front-of-pack cues on choices and eating behaviours of children and adults [85] found that children are susceptible to packaging cues, with strong visual cues the most effective, especially those using illustrations and a licensed endorser. These findings were further supported by the grey literature. Bauman et al. [86] noted in their review of opportunities for obesity prevention in 0–18-year-olds, that the removal of television advertising for energy-dense and nutrient-poor products during children’s peak viewing times would be one of the most cost-effective population-based policy measures for influencing children’s health.

The high-quality review by Von Philipson et al. [80] reported a number of effective marketing approaches to reduce discretionary choices, specifically SSBs. These included the use of traffic-light labelling on food packages, in-store promotion of healthier alternatives to SSBs and multi-component community social marketing campaigns targeting SSBs. The impact of other marketing approaches, such as the use of a nutrition rating score label on supermarket shelves and the use of calorie labelling on menus in chain restaurants, was mixed.

#### 3.4.4. Political/Policy Environment

One review [80] specifically included studies examining the impact of policy-level interventions aimed at improving the external food environment. This review by Von Philipson and colleagues [80] reported mixed findings for a number of policy-level interventions on SSB sales/consumption, including urban planning restrictions on new fast-food restaurants, industry self-regulation and trade and investment liberalisation in low- and middle-income countries.

## 4. Discussion

To our knowledge, this is the first rapid review of its kind to synthesise evidence from existing systematic reviews on the effectiveness of a wide range of interventions to improve nutrition and food sustainability across the first 2000 days of life using a broad food-systems approach. We found that most of the included systematic reviews focused on interventions targeting the behaviour of parents and caregivers, in particular breastfeeding interventions, with fewer systematic reviews focusing on other determinants of children’s diets such as the personal and external food environments. No systematic reviews were identified that focused on food supply-chain activities to get food ‘from farm to table’ despite the potential importance of this determinant in influencing children’s diets [14]. For example, a mid-way review of the 2014–2025 Scottish food policy in 2019 found positive changes to consumption behaviours through improvements along the food supply chain in addition to personal and external food environments [87]. The nature of this evidence may reflect the more traditional focus on proximal determinants of children’s diet (i.e., parents and caregivers) as being potentially easier to influence and more amenable to conventional research designs such as randomized controlled trials compared to broader systemic influences. These broader influences are harder to research due to limitations in research design and even harder to change due to the complex and interwoven nature of food systems [88].

### 4.1. Recommendations for Policy and Practice

This review highlights that there is sufficient high-quality evidence to support the integration of breastfeeding education and counselling into existing service-delivery systems with a focus on continuity of support across antenatal and postnatal services. This remains a challenge for many health-service systems, whereby antenatal care is provided by a different set of health professionals (e.g., midwives, obstetricians) in hospital settings while postnatal care is provided largely in the community by primary-care practitioners such as GPs, Child and Family Nurses/Health Visitors. Policies and programs that encourage more ‘joined up’ approaches to breastfeeding support across the antenatal and postnatal period are needed. Interventions delivered by mHealth and eHealth have potential to help provide continuity of breastfeeding support from pregnancy to postpartum; however, further high-quality research is needed to determine the effectiveness of these approaches for breastfeeding and parent nutrition interventions in general.

The findings of this review support the need for multi-component breastfeeding interventions at different levels of the socioecological model. There was evidence from low- to moderate-quality reviews to support the effectiveness of the BFHI [53,54], workplace interventions [36,52,57,58] and laws in mandating workplace lactation breaks and access to lactation support services in improving breastfeeding outcomes, including amongst minority and low-income women. Given access and uptake of individual breastfeeding support has been shown [52] to be lower amongst low socioeconomic and minority groups, these setting and policy-level interventions are likely to be particularly important in reducing the breastfeeding ‘gap’ across social, economic and cultural divides. While there is strong endorsement by the WHO for macro-level approaches, there is substantial variation across and within countries in the implementation of BFHI [89,90] and breastfeeding-related laws such as anti-discrimination laws to protect breastfeeding women, paid parental leave and workplace facilities, and paid time to support breastfeeding [91]. Implementation of BFHI is chronologically correlated with breastfeeding rates [90] and longer paid parental leave is associated with longer breastfeeding duration [92]. These structural strategies would likely achieve synergistic effects if also aligned with ensuring continuity of care across the antenatal to postnatal periods.

While the effectiveness of parent-focused interventions in improving child diet were mixed [62,63,66], more intensive interventions that directly engaged parents with interactive skill-building components were more effective than passive information provision. Evidence also suggests that including a focus on repeated exposure to vegetables and reducing access to SSBs at home are likely to be effective in improving vegetable intake [15,17,23] and reduce SSB consumption [80], respectively. As parents return to work, the ECEC setting was identified as a potentially effective setting for behavioural nutrition and social marketing interventions for both parents and children. In Australia, two-thirds of children aged 1–4 years (66.5%) attend some form of childcare, with long daycare being the most common (39%) of these [93]. Childcare educators play an important role in influencing the nutrition environment and role modelling healthy behaviours to children in their care. While studies amongst childcare educators are scarce, findings support the need for pre-service nutrition training on evidenced-based food provision and feeding practices supported by coherent centre-based nutrition policies [94,95,96]. In line with this, an Australian study found that 94% of mothers of young children supported a requirement for nutrition policies to meet a ‘best practice’ standard within childcare settings [97]. Evidence on the effectiveness of community gardens [18] suggests that ‘kitchen garden’-style programs that support repeated exposure to vegetables and the development of food literacy skills may be worthy of future research in the ECEC setting.

Evidence suggests that food supplementation [72] and voucher-style programs [73,74] along with access to local farmers markets [74] and community gardens [18] have potential for improving access and availability of healthier foods to families living in disadvantage. These schemes are more likely to be effective in improving children’s diets if they incentivise purchasing of fruit and vegetables and restrict SSB purchases. Even in high-income countries, food insecurity remains a major concern, with the United Nations reporting that in 2017–2019, 13.5%, 8.2% and 7.9% of the Australian, North America and European populations, respectively, were food insecure [98]. This is likely to be worse now given the impacts of the COVID-19 pandemic. Given the well-known impacts of food insecurity on child health, social and educational outcomes [99], these food-supplementation and voucher-style programs have an important potential role to play in improving child nutrition in the first 2000 days. It is important to note, however, that these approaches do not address the underlying determinants of food insecurity. A sustainable, stable and accessible food supply chain is a key determinant of population food and nutrition security. The *EAT-Lancet Commission on healthy diets from sustainable food systems* calls for a transformation of global food systems away from overproduction and ultra-processed outputs [13]. Stakeholders for such a transformation include those responsible for food production, storage, processing/manufacturing, distribution, packaging, retail and markets, and waste disposal [14].

There is high-quality evidence [80] to support interventions focused on improving the external food environment, including fiscal strategies such as the SSB tax, restrictions on the marketing and advertising of discretionary food and drinks, and changes to improve food labelling. Further evidence [78,100] suggests that these types of approaches are likely to have a greater impact on those most at risk of poor nutritional outcomes such as lower socioeconomic groups and hence may be particularly powerful in reducing equity gaps. Approximately 50 countries have implemented a range of SSB taxes at the city, region or country level [101], A common objection to SSB taxes is the belief that such initiatives are regressive [102], a claim often made by industry in conjunction with claims of major economic impacts [103]. Emerging evidence indicates that people experiencing higher disadvantage benefit the most from SSB taxes [104,105] and no net loss of jobs with the introduction of marketing regulation or SSB taxes [106,107,108]—yet strong opposition to such initiatives remains in some countries. This may be due to the pervasive nature of ‘nanny state’ arguments, i.e., taxes and regulation are positioned as imposing on freedom of choice [109]. This political hesitation may be influenced by public sentiment, and a meta-analysis study has found that public acceptance of SSBs is variable, depending on the narrative. For instance, a study found that while 39% of people did not support an SSB tax to reduce obesity, 66% did support the tax if the revenue was used exclusively for health initiatives [110]. Considerations of narrative are in important element in policy formation.

Governments around the world do have policy levers available to them to restrict the marketing of unhealthier foods, for example, those identified in the WHO *Recommendations on the Marketing of Foods and Non-Alcoholic Beverages to Children* [111]. However, many countries, including Australia, have a strong preference towards voluntary self-regulation of marketing practices for all foods and non-alcoholic beverages. This is despite multiple studies showing that self-regulation is not only ineffective [112] but children in countries with self-regulation are exposed to more advertising of unhealthy food on television than countries with no regulation at all [113]. There is also broad public support for advertising restrictions, with a recent study of mothers with infants finding that 92% were in support of restricting unhealthy food advertising in and around public transport [97].

Toddler food and drink products are an emerging market of concern with respect to food marketing, with the number of products in this space doubling in Australia between 2013–2018 [114]. Recent Australian research [115] showed that 30–40% of these products could be defined as discretionary choices, with 85% considered ultra-processed and 99% containing marketing messages or claims. The co-location of these products in food retail is a marketing strategy that implies a ‘natural progression’ from infant formula to follow-on milks to toddler milks, food and other drinks. One policy approach to address this would be to extend the WHO *International Code of Marketing Breastmilk Substitutes* [116] to include all commercially prepared baby and toddler ‘targeted’ food and drinks.

Overall, the evidence from this review points to the importance of action in multiple domains, from those targeting behaviour of parents/care givers, to personal and external food environments and those focusing on the food supply. These actions need to occur across a range of settings and sectors at the international, national and local levels. The global food system is influenced by international trade agreements [117], cross-border marketing and between country differences in nutrient labelling systems [91]. These agreements relate to both core foods as well as ultra-processed foods. Trade agreements have the potential to positively shape the food system, through the use of clauses identifying ultra-processed foods and their ingredients (including import volumes, tariff rates and quotas) and anti-dumping measures. At the national level, this could be supported by a national obesity prevention strategy as well as food and nutrition policy and national dietary guidelines that incorporate a focus not just on nutrition but also food systems and sustainability. Additionally, consideration should be given to increase selected agricultural subsidies (focused on whole fruit and vegetable producers, to prevent oversupply and food dumping). Evidence suggests potentially large health benefits and reduced health costs for such an initiative [118]. It is important that such policies have a funded implementation plan that incorporates a systems approach. At the local level, the use of local plans focused on early years (including promotion of nutrition), a skilled and funded workforce particular in primary health care and ECEC settings as well as local/regional food policies may help to facilitate joined-up action across settings and sectors.

### 4.2. Recommendations for Future Research

There are a number of recommendations for future research arising from this rapid review. From a methodological perspective, future systematic reviews should not only adhere to the updated PRISMA reporting guidance for systematic review [119], but the planning and conduct of the review should be informed by the AMSTAR 2 assessment tool [31]. Our finding that less than one in five systematic reviews included in this rapid review were of high quality underscores the importance of addressing methodological weaknesses in future systematic reviews in this area. Key areas of weaknesses (i.e., more than 50% of reviews) identified in the quality assessment undertaken in this study relate to item 7 and item 10 of the AMSTAR 2 assessment tool. Item 7 (a critical domain) requires authors to provide a list of potentially relevant studies excluded from the review along with a justification for each exclusion. This is important, as unjustified exclusion of studies has the potential to bias the review findings [31]. Item 10 (noncritical domain) requires studies to report on the sources of funding for studies included in the review given that the results of industry funded studies sometimes favoured sponsored products or interventions.

This rapid review also identifies a number of areas for future research and evidence synthesis to build the evidence base for what works to improve nutrition and food sustainability in the first 2000 days of life. A key gap identified for future research synthesis is the influence of food-supply issues on the diets of both adults and young children, and food-supply interventions aimed to improve both nutrition and food sustainability. This should include an examination of both systematic reviews and individual studies that examine impacts beyond the first 2000 days. Such research is pertinent, given the known importance of a sustainable food system [13] to tackling complex issues such as population-level food and nutrition security and climate change. This could be supplemented with qualitative research to investigate stakeholders’ perspectives for aligning the food sector, in particular agriculture and trade, including nutrition aims for healthy and sustainable diets focused on the first 2000 days.

### 4.3. Review Strengths and Limitations

Rapid-review methods are modified to generate evidence in a short time [120]; however, short timeframes may lead to inadequate reporting and this can be a limitation of rapid reviews [121]. Whilst every effort has been made to approximate a full systematic review, and a thorough search was undertaken consistent with the agreed scope of work, it is possible that some relevant studies were missed. One limitation of systematic reviews is that they are not necessarily contemporary, and thus more recent individual studies may not be represented in this review. It is important to note that evaluating methodological or evidence quality using the AMSTAR 2 tool is a subjective process and although the included systematic reviews and meta-analyses have been evaluated independently by two researchers, there may still be some bias [122]. Finally, it is important to acknowledge that this report does not include a full representation of the grey literature, as this was well beyond scope. The review has sought, however, to utilise sentinel grey literature to provide additional information and perspective on any given domain.

## 5. Conclusions

Most of the systematic review evidence on nutrition in the first 2000 days of life focuses on interventions targeting the behaviour of parents and caregivers, in particular breastfeeding interventions, with fewer systematic reviews focusing on other determinants of children’s diets such as the personal and external food environments. A key gap identified in the evidence base is the influence of food-supply issues on the diets of both adults and young children and food-supply interventions aimed at improving both nutrition and food sustainability. Evidence supports the integration of multi-component breastfeeding interventions at different levels of the socioecological model into existing service-delivery systems and policy. While the effectiveness of parent-focused interventions in improving child diet were mixed, more intensive interventions that directly engaged parents with interactive skill-building components were more effective than passive information provision. Evidence also suggests that including a focus on repeated exposure to fruit and vegetables at both home and childcare is effective in increasing fruit and vegetable intake. Our findings support the use of food vouchers subsidizing the cost of healthier foods, community gardens and farmers markers in improving availability and access to nutritious foods. There is high-quality evidence to support interventions focused on improving the external food environment, including restrictions on the marketing and advertising of discretionary food and drinks, fiscal strategies such as the SSB tax and changes to improve food labelling. Overall, this review highlights the importance of action across a range of sectors and settings at the international, national and local levels to improve young children’s diets.

## Figures and Tables

**Figure 1 nutrients-14-00731-f001:**
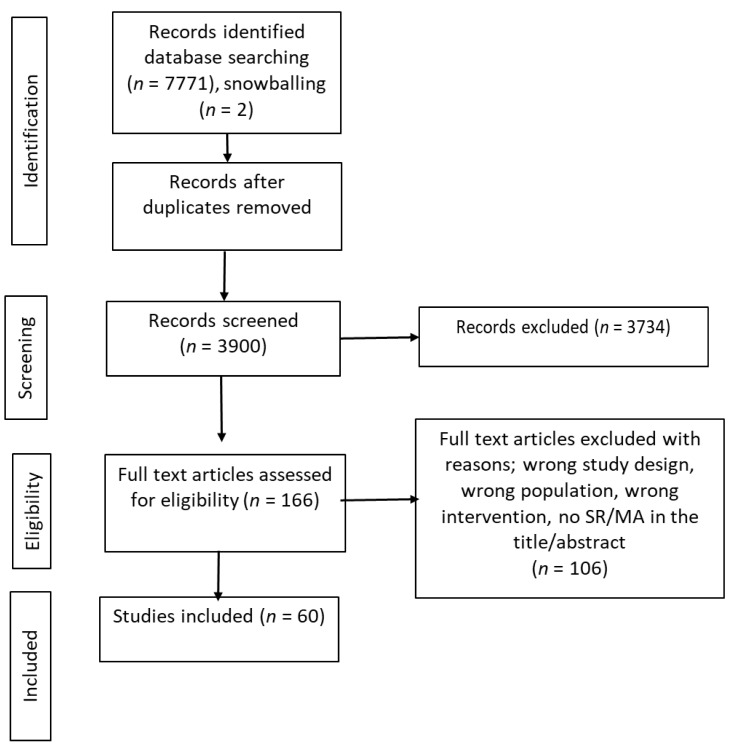
Preferred Reporting Items for Systematic Reviews and Meta-analyses (PRISMA) flow diagram.

**Table 1 nutrients-14-00731-t001:** Search Strategy: PICOTT components and application to this review.

PICOTT Components	Application to This Review
P—Patient, population or problem	Pregnancy, infants and children aged 0–5 years, broader food environments, high income countries.
I—Intervention, exposure, prognostic factor	Any interventions that aims to improve food, nutrition or food sustainability in line with the Innocenti domains (food supply chains, external food environments, personal environments, and caregiver be-haviour).
C—Comparison	No intervention or usual care/practice
O—Outcome	Improvements to food, nutrition and food sustainability
T—Type of question	Prevention
T—Type of study	Systematic review or meta-analyses of randomised trials, non randomised trials or of longitudinal studies plus grey literature reports

**Table 2 nutrients-14-00731-t002:** Overview of the focus and quality of the reviews mapped to Innocenti Framework.

Innocenti Framework Determinants	Intervention Focus of Reviews	Systematic Reviews and Meta-Analyses (with AMSTAR 2 Rating)					
		High	Moderate	Low	Critically Low	Total Reviews	Grey Literature
Behaviour of caregivers and children	Breastfeeding Program/peer support	2	0	7	7	16	0
	Breastfeeding mHealth/eHealth/telehealth	2	0	0	1	3	0
	Breastfeeding intervention for vulnerable/high risk groups	2	0	2	2	6	0
	Breastfeeding in health services	0	0	0	4	4	0
	Breastfeeding in workplaces	0	0	0	2	2	2
	Parent nutrition interventions	1	0	6	4	11	1
	Fruit and vegetable consumption	2	0	2	1	5	0
	Early Childhood Education Care Setting	0	0	0	2	2	7
	**Sub Total**	**9**	**0**	**17**	**23**	**49**	**10**

Personal Food Environments	Food supplementation/vouchers	1	0	0	2	3	5
	Fruit and vegetables	0	0	1	0	1	0
	**Sub Total**	**1**	**0**	**1**	**2**	**4**	**5**

External Food Environments	Sugar sweetened beverages	1	0	1	0	2	5
	Discretionary choices	0	0	0	1	1	0
	Marketing/advertising	0	0	2	2	4	1
	**Sub Total**	**1**	**0**	**3**	**3**	**7**	**6**
Food Supply Chains		0	0	0	0	0	2
**TOTAL**		**11**	**0**	**21**	**28**	**60**	**23**

**Table 3 nutrients-14-00731-t003:** Behaviour of caregivers and children domain: Summary of intervention evidence.

Intervention	Evidence	No. Review ^1^—Quality
**Breastfeeding programs/peer support**		
Antenatal breastfeeding education to increase breastfeeding duration		1 review—high [33]
Education, peer/health professional counselling to promote initiation of breastfeeding		1 review—high [34] 2 reviews—critically low [35,36]
Education and support to promote exclusive breastfeeding		2 reviews—low [19,37] 1 review—critically low [36]
Peer counselling and support for promoting breastfeeding initiation, duration and exclusivity		6 reviews—2 low [38,39], 4 critically low [35,36,40,41]
Targeting father/partners for breastfeeding promotion.		1 review—low [42]
Mother-infant skin-to-skin contact to promote exclusive breastfeeding		1 review—critically low [43]
Postnatal face-to-face contact with a health professional to promote breastfeeding duration and exclusivity		1 review—critically low [44]
**Breastfeeding mHealth/eHealth/telehealth**		
Telephone support (mainly voice calls) during pregnancy and early post-partum for breastfeeding		1 review—high [45] 1 grey literature review
Targeted client communication via mobile device (e.g., SMS, voice calls, apps with instant messaging) for breastfeeding		1 review—high [46]
Internet support for breastfeeding		1 review—critically low [47]
**Breastfeeding interventions for priority population groups**		
Education and counselling for adolescents		1 review—low [48]
Education and support for overweight or obese women		1 review—high [49]
Peer counselling, environmental supports for low income women (USA Women Infant Children Program)		1 review—critically low [50]
Breastfeeding education and support for minority women		2 reviews—1 critically low [51] 1 moderate [52]
Pregnancy focused intervention in Indigenous women		1 review- low [28]
Macrosystem/policy level interventions for minority women		1 review—moderate [52]
Breastfeeding friendly maternity care practices in hospitals for minority women		1 review—moderate [52]
**Breastfeeding in Health Services**		
Breastfeeding Friendly Hospital Initiative (BFHI)		2 reviews—critically low [53,54]
Education and training for healthcare staff		2 reviews—critically low [55,56]
**Breastfeeding in the Workplace**		
Employer based breastfeeding programs		3 reviews–all critically low [36,57,58]
**Parent nutrition interventions**		
Nutrition education in pregnancy		3 reviews—1 low [59] and 2 critically low [60,61]
Pregnancy focused intervention for Indigenous women		1 review—low [28]
Parent focused nutrition interventions		6 reviews—1 high [62] 3 low [59,63,64] and 2 critically low [65,66]
Digital nutrition interventions		1 grey literature review [67]
**Fruit and vegetable consumption**		
Nutrition education alone		3 reviews—2 high [15,68], 1 low [17]
Fruit and vegetable tasting (repeated exposure) at both home and ECEC		4 reviews—2 high [15,68], 2 low [17,23])
Multi-component interventions (including ECEC)		3 reviews (2 high [15,68], 1 critically low [16])
**Early Childhood Education and Care Setting (ECEC)**		
Behavioural intervention		2 reviews—1 low [69] and 1 critically low [22])
Social marketing		1 review (critically low) [21]

^1^ Some reviews are counted under more than one intervention type depending on the focus of the review. □ Supportive evidence (high confidence)—1 or more reviews of high quality that concluded that interventions were effective. □ Supportive evidence (low confidence)—1 or more reviews of moderate, low or critically low quality that concluded that interventions were effective. □ Mixed evidence—1 or more reviews of any quality that concluded that the evidence for effectiveness was mixed (some positive studies, some negative studies). □ Insufficient or limited evidence—1 or more reviews of any quality that concluded that there was insufficient or limited evidence of the effectiveness of interventions.

**Table 4 nutrients-14-00731-t004:** Personal Food Environments: Summary of intervention evidence.

Intervention Focus	Evidence	No. Review ^1^—Quality
Food supplementation		1 review—high [72]
Food vouchers		2 reviews—critically low [73,74]
Increasing availability and accessibility of farmers markets		1 review—critically low [74]
Gardening interventions		1 review—low [18]

^1^ Some reviews are counted under more than one intervention type depending on the focus of the review. □ Supportive evidence (high confidence)—1 or more reviews of high quality that concluded that interventions were effective. □ Supportive evidence (low confidence)—1 or more reviews of moderate, low or critically low quality that concluded that interventions were effective.

**Table 5 nutrients-14-00731-t005:** External Food Environments: Summary of Intervention Evidence.

Interventions	Evidence	No. Review ^1^—Quality
**Physical food environment**		
Home based interventions improving availability of healthier alternatives to SSB at home ^2^		1 review high [80]
Reducing young children physical access to SSB and increase access to healthy beverages		1 review—low [81]
Improving the school food environment- reduced availability of SSB, improved access to water, fruit and healthier vending machines ^2^		1 review—high [80]
Healthier default beverages on children’s menus in chain restaurants ^2^		1 review—high [80]
**Fiscal/economic environment**		
SSB tax/Price increase ^2^		1 review high [80]
Discretionary food tax ^2^		1 review—high [80]
Food voucher schemes with incentive for purchasing fruit and vegetables and restrictions on SSB purchases ^2^		1 review—high [80]
Food voucher schemes without SSB restriction ^2^		1 review—high [80]
Price discount on low calorie beverages via supermarket loyalty cards ^2^		1 review—high [80]
**Marketing and Advertising**		
Eliminate advertising of SSB and discretionary foods in public places ^2^		1 review—high [80]
Reduce screen and other marketing to children		1 review—critically low [82] 1 review—low [83]
Media character marketing could be used to support healthy food environments for children		1 review—low [84]
Front of pack cues on food packages		1 review—low [85]
Multi-component community campaigns targeting SSB ^2^		1 review—high [80]
In store promotion of healthier alternatives to SSBs ^2^		1 review—high [80]
Traffic light labelling on food packages^2^		1 review—high [80]
Nutritional rating score label on supermarket shelf ^2^		1 review—high [80]
Menu board calorie labelling in chain restaurants ^2^		1 review—high [80]
**Political/policy environment**		
Urban planning restriction on new fast food restaurants ^2^		1 review—high [80]
industry self-regulation to improve nutrition quality of whole food supply ^2^		1 review—high [80]
trade and investment liberization in low and middle income countries ^2^		1 review—high [80]
Restrictions to number of stores selling SSB in remote communities ^2^		1 review—high [80]

^1^ Some reviews are counted under more than one intervention type depending on the focus of the review. ^2^ This review specifically focused on the effect of these interventions on SSB sales/consumption. □ Supportive evidence (high confidence)—1 or more reviews of high quality that concluded that interventions were effective. □ Supportive evidence (low confidence)—1 or more reviews of moderate, low or critically low quality that concluded that interventions were effective. □ Mixed evidence—1 or more reviews of any quality that concluded that the evidence for effectiveness was mixed (some positive studies, some negative studies).

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
