# Peer review of "What Works to Improve Nutrition and Food Sustainability across the First 2000 Days of Life: A Rapid Review"

_nutrients, 2022, doi:10.3390/nu14040731_

Round 1

Reviewer 1 Report

The results of this quick review should provide extensive guidance for practitioners and policy makers on "best buys" and areas in which to invest to promote healthy and sustainable diets during the first 2000 days of life.
The review proposed by the authors is very interesting, but I suggest to include summary tables of the results.
The use of tables makes understanding the results "rapid".
I recommend to include the authors' conclusions.
The period is missing from line 68.

Reviewer 2 Report

Thank you very much for allowing me to review the review article titled “nutrients-1541971_What works to improve nutrition and food sustainability across the first 2000 days of life: A rapid review.”

This is a review presented to Nutrients for the Section “Nutrition and Public Health”.

I have read this excellent work with enthusiasm. I consider that it has a suitable approach and integrates all the most outstanding knowledge of the last years.

The applied methodology is correct. The assessment of the works is correct; the discussion raises an outstanding interest in the future based on current knowledge.

This article is very well written.

Round 2

Reviewer 1 Report

I have no other comments. Good job!